# Accelerating Pre-training of Multimodal LLMs via Chain-of-Sight

Ziyuan Huang[1]   Kaixiang Ji[1]   Biao Gong[1]   Zhiwu Qing[2]   Qinglong Zhang[1]
Kecheng Zheng[1]   Jian Wang[1]   Jingdong Chen[1]   Ming Yang[1]
[1]Ant Group   [2] Huazhong University of Science and Technology
https://chain-of-sight.github.io/

## Abstract

This paper introduces Chain-of-Sight, a vision-language bridge module that accelerates the pre-training of Multimodal Large Language Models (MLLMs). Our approach employs a sequence of visual resamplers that capture visual details at various spacial scales. This architecture not only leverages global and local visual contexts effectively, but also facilitates the flexible extension of visual tokens through a compound token scaling strategy, allowing up to a $16\times$ increase in the token count post pre-training. Consequently, Chain-of-Sight requires significantly fewer visual tokens in the pre-training phase compared to the fine-tuning phase. This intentional reduction of visual tokens during pre-training notably accelerates the pre-training process, cutting down the wall-clock training time by $\sim$**73%**. Empirical results on a series of vision-language benchmarks reveal that the pre-train acceleration through Chain-of-Sight is achieved without sacrificing performance, matching or surpassing the standard pipeline of utilizing all visual tokens throughout the entire training process. Further scaling up the number of visual tokens for pre-training leads to stronger performances, competitive to existing approaches in a series of benchmarks.

## 1   Introduction

Recently, Large Language Models [70, 6, 80, 5, 3] have received unprecedented attention, owing to their remarkable capabilities in text comprehension and generation. Riding on the success of LLMs, Multimodal Large Language Models (MLLMs) [74, 90, 63, 56, 26, 88, 73] demonstrate impressive zero-shot transferability across a wide range of vision-language tasks, such as image captioning, visual question answering, and visual grounding.

The exceptional generalization ability exhibited by the contemporary MLLMs can be largely attributed to their extensive pre-training on a massive amount of data [16, 14, 69, 31, 18]. However, as the volume of data escalates, so does the wall-clock training time, which has become a major obstacle in further explorations. According to [63], 60,000 GPU hours are needed for training a 7B model on just 96 million image-text pairs. This intensive computational demand is not only prohibitive to many researchers, but also leads to a significant carbon footprint.

One of the key reasons for the prolonged training time is the extensive length of visual tokens Typically, the image-text pairs in the pre-training phase involve around 23 text tokens (see Table 1). In contrast, most MLLMs handle substantially more visual tokens during pre-training, *e.g.*, 144 [10, 11], 256 [4, 50, 97], or even higher [63, 16, 55, 56, 64, 48]. Reducing the number of visual tokens presents a straightforward way to speed up training, as it allows for an increase in batch size and a concurrent decrease in step time. Meanwhile, the reduced memory consumption allows for better optimization stages [81], further reducing time requirements. However, training with fewer visual tokens often results in compromised performance for existing vision-language models.

38th Conference on Neural Information Processing Systems (NeurIPS 2024).

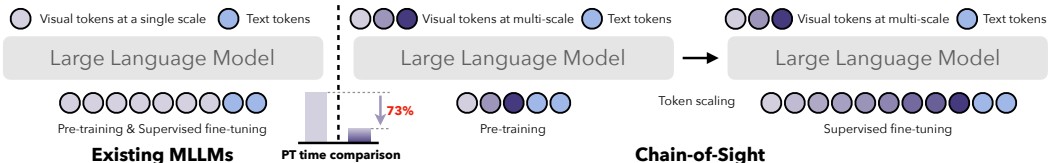

Figure 1: **Chain-of-Sight concept overview**. Recent current MLLMs maintain a constant set of visual tokens in both pre-training and fine-tuning. These tokens typically represent visual contents at a single visual scale. In contrast, our Chain-of-Sight approach leverages the idea of visual hierarchy, producing multi-scale visual tokens. Moreover, the token scaling strategy enabled by our multi-scale visual resamplers allow us to start with a small pool of visual tokens for pre-training, before increasing the number of tokens during fine-tuning. This considerably accelerates the pre-training phase.

To resolve this dilemma, this work introduces Chain-of-Sight, a vision-language bridging module for efficient pre-training of MLLMs. Unlike existing approaches that maintain a constant token count throughout both pre-training and fine-tuning, Chain-of-Sight allows for a marked increase in the number of tokens after the pre-training stage, thereby reducing the tokens needed during pre-training. The core mechanism is our multi-scale visual resampler, which produces visual tokens of multiple visual scales. Inspired by the classical concept of multi-scale feature hierarchy in visual understanding [106, 42, 33, 107, 83, 53], we partition the visual features produced by the visual backbone using windows of multiple sizes. For each window size, a visual resampler is implemented to produce a specified number of visual tokens per window. Subsequently, the visual tokens from various window sizes are gathered and linked in a global-to-local manner, forming a chain of reasoning steps from coarse views gradually to fine-grained perspectives.

On top of this, we propose a post-pretrain token scaling strategy, which compounds the elements of input resolution and window size manipulation to enable a significant escalation in the token count for our Chain-of-Sight, reaching up to $16\times$ increase during fine-tuning. Such adaptability allows for the fine-tuning of the model with a flexible granularity or complexity as required, without the the necessity for an additional pre-training phase.

By intentionally reducing the number of visual token by $\sim$90% in the pre-training, a $2.5\times$ batch size is allowed with a step time reduction of 30%, leading to a $3.7\times$ faster pre-training in terms of wall-clock time ($\sim$73% less) for the same amount of data, when compared with using all visual tokens during pre-training. Meanwhile, our observations indicate that this acceleration does not come at the expense of performance. The results achieved by our Chain-of-Sight model pre-trained with 32 tokens match or surpass those obtained using 336 visual tokens throughout the training process, when both models use the same tokens during fine-tuning. Further scaling up the tokens in the fine-tuning stage leads to enhanced performance at small additional training costs. This scaling showcases the potential of Chain-of-Sight to capitalize on the initial efficiency gains and adapt its framework to achieve even greater levels of accuracy and effectiveness in visual understanding for MLLMs.

## 2    Method

Our objective is to accelerate the pre-training of MLLMs. To this end, we resort to reducing the number of visual tokens inputted into the language model. To mitigate the performance drop associated with fewer visual tokens, we introduce a versatile bridge module within our framework, named Chain-of-Sight. This module is designed to enable the increase in the token count on demand after pre-training. With this capability, we are able to substantially lower the number of visual tokens during the pre-training phase, while retaining the ability to scale up and capture a rich level of visual detail during fine-tuning. The concept of Chain-of-Sight is illustrated in Fig. 1.

### 2.1    Re-examining the efficiency bottleneck in MLLM pre-training

Modern MLLMs are typically constructed by three core components: (1) a visual encoder, (2) a vision-language bridge module, and (3) a language model. Given that the language models often have a much larger size than the visual encoder, they account for the majority of computation during pre-training [90, 4, 73, 19, 58]. Consequently, the number of input tokens processed by the language model is a crucial factor determining the total computational workload.

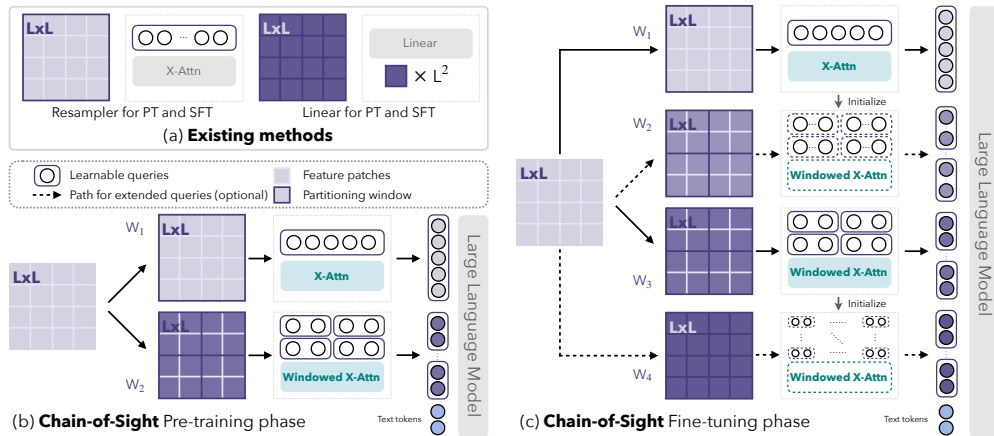

Figure 2: **The Chain-of-Sight framework.** Through partitioning visual features into windows and restricting cross-attention to the windowed features associated with the learnable tokens, our Chain-of-Sight approach produces visual tokens that encompass multiple scales. Thanks to the post-pretrain token scaling strategy, Chain-of-Sight reduces the required number of visual tokens in pre-training, thus accelerating the process. In contrast, the number of visual tokens remains constant in resampler-based methods [44, 2, 4, 99] for pre-training and fine-tuning, and the linear-layer [56, 63, 97, 15] produce a large number of visual tokens, incurring a high cost for pre-training.

As detailed in Table 1, the pre-training data predominantly comprise image-text pairs that contain fewer than 50 text tokens. In contrast, existing MLLMs are designed to handle $2\times$ more visual tokens, often requiring such as 144 [10, 11], 256 [4, 50, 97], or even more visual tokens [63, 16, 55, 56, 64, 48]. The imbalance between the visual tokens and text tokens means that processing these visual tokens has become the main efficiency bottleneck in MLLM pre-training. This prompts our exploration for more efficient vision-language bridging structures, which is capable of reducing the number of visual tokens in pre-training without compromising performance.

## 2.2 Multi-scale visual resamplers

The multi-scale visual resamplers serve as the foundational mechanism enabling the flexible extension of visual tokens after the pre-training phase. This subsection focuses on the architectural details of the multi-scale visual resamplers, as visualized in Fig. 2(b), while the the extension of visual tokens is discussed in the subsequent subsection.

Essentially, the idea of exploiting multi-scale or pyramid structures to handle natural hierarchy of visual contents has been long established as a standard practice [30, 42], proving effective in countless visual tasks [33, 61, 83, 53]. Despite this, the potential for harnessing multi-scale visual hierarchies remains under-explored in the context of MLLMs.

**Visual resampler.** Visual resampler is a Perceiver [36]-like structure that introduces a set of learnable queries and uses cross-attention to condense visual knowledge into a predetermined set of visual tokens [4, 2, 95, 99]. We construct Chain-of-Sight with visual resamplers due to their flexibility in selecting the token count for a specified feature, independent of the features' dimensionality.

**Multi-scale visual resamplers.** One of the effective strategies for building multi-scale features within a network involves combining operations that spans diverse fields of views [13, 45, 101, 21]. Given that the resampler structure inherently gathers visual cues on a global scale across the entire feature map, our strategy focuses on enhancing the perception of the fine-details in the image.

To this end, we partition the visual features into non-overlapping local windows of various sizes. More precisely, given a visual feature $\mathbf{X} \in \mathbb{R}^{L \times L \times C}$ extracted by the visual encoder, where $L$ and $C$ denote the feature size and channel, respectively, we define a set of window sizes, denoted as $\mathbf{W} = [W_1, ..., W_m]$. This setup leads to a collection of windowed visual features $\mathbf{X}_{\text{win}} = [\mathbf{X}_1, ..., \mathbf{X}_m]$. Each $\mathbf{X}_i$ represent a set of $L^2/W_i^2$ windowed features obtained by applying the partition operation on the original visual feature maps with a corresponding window size $W_i$. This naturally forms features of multiple visual scales.

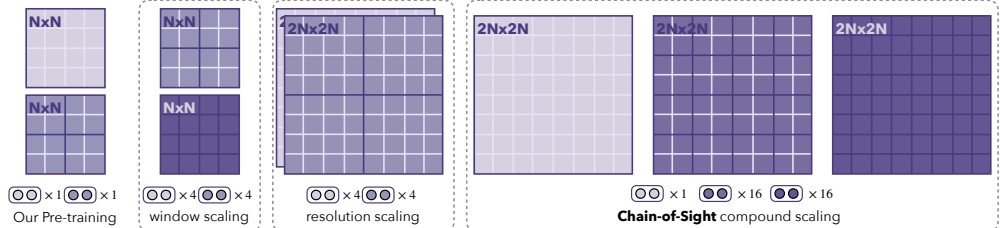

Figure 3: **Detailed illustration of our post-pretrain token scaling strategy.**

At every scale level, each windowed feature is allocated with $N_i$ learnable queries. These learnable queries are then utilized within the visual resampler to perform cross-attention solely on their corresponding windowed feature. This yields $N$ tokens, where the $N$ can be calculated as follows:

$$N = \sum_i L^2 / W_i^2 * N_i. \tag{1}$$

The learnable queries within the same scale share the parameters of the visual resampler, despite their different spatial locations. However, because the queries at various scales are intended to capture features from varying fields of view, distinct sets of parameters are used for each scale. This results in a group of visual resamplers operating across multiple scales. On top of this, we enable the resamplers to aggregate visual features from multiple feature levels, as in [8] (see Appendix for details).

**Coarse-to-fine integration.** Upon acquiring a series of multi-scale visual tokens from the multi-scale visual resamplers, our method integrates these prompts in a structured coarse-to-fine fashion. The final token sequence fed to the language model begins with tokens derived from larger windows, which presents an overall view of the image, and proceeds with tokens obtained from smaller windows that contains fine-grained details. Our priliminary experiments reveal a substantial difference in the overall performance between the coarse-to-fine and the reversed order.

## 2.3   Post-pretrain token scaling strategy

Reducing the number of visual tokens can effectively accelerate pre-training, but typically at the expense of performance. To address this dilemma, we enhance the token count after pre-training, which allows accelerated pre-training with fewer visual tokens, while a subsequent increase in tokens ensures the final performance after fine-tuning, as demonstrated in Fig. 2(c). Specifically, based on the multi-scale visual resamplers, the increase in the token count is accomplished via our compound token scaling strategy that integrates two core mechanisms: resolution scaling and window scaling.

**Resolution scaling.** Enhancing the input resolution stands as the most direct way to augment the number of visual tokens. At the cost of additional computation overhead in the visual backbone, it allows for a quadratic rise of the token count with the resolution enhancement. The concept of resolution scaling is investigated in many existing approaches based on linear projectors [55, 69, 26] or visual resamplers [50]. They can broadly be viewed as particular instances within our Chain-of-Sight framework, which regards the window size as a fixed factor. In this context, linear projectors use the smallest possible window size for visual token generation, whereas visual resamplers employ the resolution in the pre-training phase as their window size.

**Window scaling.** The windowing mechanism in our multi-scale visual resamplers enable scaling up token numbers by manipulating the window sizes. As in Eq. 1, reducing the window sizes can further produce a quadratic increase in the number of visual tokens on top of the resolution enhancement.

**Compound scaling.** Combining the above token scaling strategies, our compound scaling is capable of producing a $16\times$ increase in the tokens during fine-tuning, as in Fig. 3. This allows us to fine-tune the scale at which visual features are represented and sampled, improving the model's capability of leveraging varying levels of detail and abstraction inherent in the visual content. Consequently, the Chain-of-Sight framework significantly boosts the visual comprehension capability of the model during the fine-tuning stage, effectively compensating for the performance drop incurred by the low number of visual tokens during pre-training.

**Initialization.** Inspired by [9], we initialize the parameters of the newly introduced visual resamplers by simply inflating the pre-trained parameters, as in Fig. 2. As for the new visual queries, we apply a nearest neighbor strategy to initialize them based on the pre-trained queries.

# 3 Experiments

In this section, we provide our experimental setup, empirical results, and the comparisons with existing methods.

## 3.1 Experimental setup

**Model details.** We instantiate our MLLM with CLIP-ViT-L/14 [78] as the visual encoder and Vicuna [20] as the language model. For efficiency, we adapt Vicuna with LoRA [34] during all training stages, instead of fully fine-tuning the language model. For the number of visual tokens, we experimented with 32, 48, and 80 during pre-training for our Chain-of-Sight model, where 16 tokens are global tokens (with a window size of 16 for an input resolution of 224) and the rest are local tokens (with a window size of 4 by default). These models are configured to be extended to at most 528, 784, and 1296 visual tokens during fine-tuning using compound token scaling.

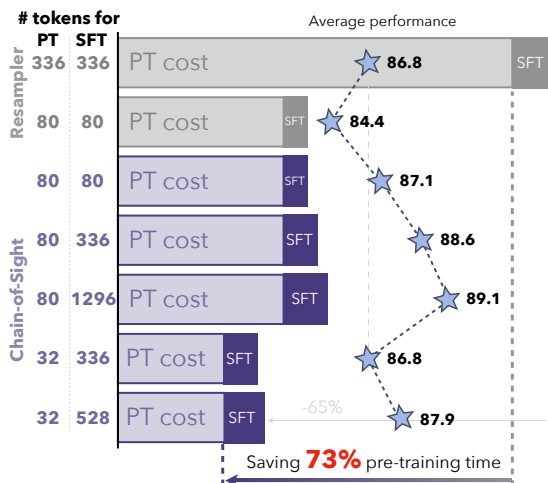

Figure 4: **Pre-train acceleration by Chain-of-Sight**, in comparison with standard resamplers. The average performance is computed over the reported benchmarks in Table 2. Our method achieves a pre-train acceleration of 73% without compromising performance.

**Training settings.** The training of Chain-of-Sight is divided into two stages. For the first stage, we sample around 65M image-text data involving multiple tasks, as detailed in Table 1. The multi-scale visual resamplers and the LoRA paramters [34] are unlockded for training. For the first 120,000 iterations, we use the input resolution of 224 and unlock the resamplers and the LoRA parameters [34] for training. During the last 30,000 iterations, the input resolution is raised to 448, where the parameters in the visual backbone is further activated and the tokens are scaled up through our compound scaling. The second stage of the Chain-of-Sight model is supervised fine-tuning, where we remove all the captioning datasets except for COCO.

**Evaluation benchmarks.** The evaluation of our approach involves various tasks including image captioning, visual question answering, text recognition, as well as the tasks defined in popular vision-language benchmarks. Details can be seen in Table A2.

## 3.2 Ablations

We first ablate the Chain-of-Sight (CoS) design for accelerating the pre-training of MLLMs. For the ablations, we omit the high resolution tuning in the first stage unless otherwise specified.

**Pre-train acceleration by Chain-of-Sight.** Fig. 4 shows the cost for pre-training and supervised-finetuning with various number of visual tokens, as well as the corresponding average performance over 12 benchmarks. We make several key findings. (a) Though reducing visual tokens for the resampler from 336 to 80 significantly reduces the training time, the average performance drops from 86.8 to 84.4. (b) Using an identical number of tokens, *i.e.,* 80 visual tokens, Chain-of-Sight notably outperforms the standard resamplers, which can be mainly accredited to the multi-scale visual tokens generated by our method. (c) Using the pre-trained model with 80 visual tokens, Chain-of-Sight can be fine-tuned with higher token counts. Using 336 tokens for fine-tuning, our method achieve an average improvement of 1.8pt over the standard resampler with 336 tokens. (d) Notably, Chain-of-Sight with 32 tokens can save up to 73% of the pre-training time, and maintain the same performance as the standard resampler with 336 tokens. (e) Taking fine-tuning into consideration, our method is capable of saving 65% of the total wall-clock time for training a MLLM with improved performance. Note that the percentage is based on a 65M pre-training dataset, and the overall gains in efficiency are expected to grow with the increase of the pre-training dataset scale.

**Image captioning, visual question answering, and text recognition.** Table 2 compares the performance of Chain-of-Sight with its baselines. Overall, our method delivers competitive performance against pre-training with a full set of visual tokens, while substantially accelerating training speed.

Table 1: **Multitask pretraining data** for pre-training Chain-of-Sight. MeanL., 50%L., and 90%L. indicates the mean length, the 50 percentile, and the 90 percentile length of the input text tokens. We use the tokenizer from Vicuna [20], which is the same tokenizer we use for pre-training.

| Task | MeanL. | 50%L. | 90%L. | Dataset |
|---|---|---|---|---|
| Caption | 23.87 | 19 | 43 | COYO [7], CC3M&12M [85], COCO [54], VG Cap [39], SBU [75] |
| General VQA | 12.17 | 11 | 17 | VQAv2 [32], GQA [35] |
| KB QA | 38.67 | 39 | 48 | OK-VQA [68], AOK-VQA [84], ScienceQA [65] |
| Text | 15.32 | 13 | 25 | TextVQA [87], OCRVQA [71], TextCaps [86] |
| REC | 12.65 | 16 | 19 | RefCOCO [38], RefCOCO+ [103], RefCOCOg [67] |
| Total | 23.32 | 19 | 42 | |

Table 2: **Image captioning, visual question answering, text recognition, and vision-language benchmarks**, compared with our baselines. † indicates fine-tuning with 224×224 resolution. ∗ denotes token extended through existing strategies [57, 50, 55]. S-I denotes the image subset of SEEDBench [43]. The best and second-best performances are marked **bold** and underlined.

| Bridge | pre-train tokens | time | FT tokens | Captioning COCO | Flickr | NoCaps | VQA v2 | OK | GQA | SQA | Text Caps | VQA | VLM-Bench MMB | POPE | S-I |
|---|---|---|---|---|---|---|---|---|---|---|---|---|---|---|---|
| *224 resolution fine-tuning* | | | | | | | | | | | | | | | |
| Linear | 256 | 0.82× | 256† | 140.1 | 78.6 | 116.7 | 79.7 | 57.3 | 62.2 | 90.2 | 120.0 | 48.7 | 67.9 | 83.2 | 66.3 |
| CoS | 80 | 0.42× | 80† | 139.6 | 78.0 | 115.8 | 79.0 | 58.0 | 61.6 | 90.4 | 120.8 | 48.3 | 68.6 | 84.4 | 64.6 |
| | 80 | 0.42× | 336† | 140.8 | 80.0 | 117.2 | 79.8 | 58.4 | 62.3 | 90.0 | 122.6 | 50.2 | 69.2 | 84.8 | 65.9 |
| *448 resolution fine-tuning* | | | | | | | | | | | | | | | |
| Resamp. | 336 | 1.00× | 336 | 141.2 | 81.9 | 117.2 | 81.0 | 58.4 | 61.9 | 84.3 | 135.5 | 61.3 | 66.9 | 85.3 | 66.2 |
| | 80 | 0.42× | 80 | 138.8 | 81.4 | 115.8 | 80.3 | 58.0 | 61.6 | 84.5 | 115.8 | 59.0 | 68.1 | 84.5 | 64.4 |
| | 80 | 0.42× | 400* | 139.8 | 84.2 | 117.3 | 81.3 | 58.3 | 61.9 | 87.3 | 135.7 | 61.9 | 68.6 | 86.0 | 66.3 |
| CoS | 336 | 1.00× | 336 | 141.4 | 83.4 | 116.8 | 80.9 | 58.4 | 62.8 | 88.6 | 133.9 | 60.3 | 68.7 | 84.2 | 66.9 |
| | 80 | 0.42× | 80 | 140.7 | 82.6 | 118.0 | 80.7 | 58.4 | 61.6 | 89.6 | 134.5 | 60.4 | 69.0 | 84.5 | 65.0 |
| | 80 | 0.42× | 336 | 141.3 | **85.8** | 119.2 | 81.7 | 58.9 | 62.1 | 90.5 | 137.8 | 63.8 | 70.2 | 85.9 | 66.4 |
| | 80 | 0.42× | 1296 | **142.8** | 84.9 | **119.3** | 82.5 | 59.4 | 62.5 | 91.5 | 137.5 | **65.0** | **70.3** | **86.4** | 67.5 |
| *further accelerations* | | | | | | | | | | | | | | | |
| CoS | 48 | 0.35× | 784 | 142.7 | 83.4 | 119.1 | 82.3 | 59.1 | **62.7** | 91.0 | **138.4** | 64.7 | 69.7 | 84.3 | 67.3 |
| | 32 | 0.27× | 336 | 141.4 | 83.5 | 118.1 | 80.7 | 57.8 | 61.1 | 89.3 | 133.7 | 59.5 | 66.8 | 84.0 | 65.1 |
| | 32 | 0.27× | 528 | 141.7 | 83.1 | 117.1 | 81.6 | 58.3 | 62.4 | 91.1 | 136.4 | 61.8 | 69.4 | 85.3 | 66.7 |

Compared to the linear projection, we fine-tune the visual backbone at a resolution of 224x224. Pre-trained with 80 tokens, CoS slightly falls behind when we use 80 tokens for fine-tuning, while achieving stronger performance when we scale up the tokens to 336. In this case, the total time for pre-training can be saved by 50% when compared to using linear projections with 256 visual tokens.

Compared with visual resamplers, when a high number of visual tokens are used during pre-training, Chain-of-Sight performs similarly. However, when pre-trained with fewer visual tokens, Chain-of-Sight have notable advantages on captioning and text recognition for 80 tokens, outperforming resamplers by 1.8 for captioning and 10.1 for text recognition. Further scaling up tokens pushes our model to have a performance stronger than the model pre-trained with 336 tokens. In addition, our compound scaling proves more effective than existing resolution scaling methods [57, 50, 69].

We also try further reducing the number of tokens for pre-training. The performance drop is mainly observed in vision-language benchmarks and question answering capabilities, with minor affect on the captioning task. Nevertheless, we observe that our model pre-trained with 32 tokens achieve a similar or stronger overall performance with resamplers pre-trained with 336 tokens, while taking only 0.27× wall-clock training time (achieving a 3.7× pre-train acceleration).

**Vision-language benchmarks.** We find the performances on vision-language benchmarks more heaviliy affected by the number of tokens used in the supervised fine-tuning stages than the ones used in the pre-training stage, especially for the question answering task with multiple choices, as in Table 2. This provides a strong support for using a small set of visual tokens in the pre-training stage for acceleration, and a large set of visual tokens for optimal performance.

Table 3: **Ablations on referring expression comprehension** compared with our baselines. † indicates fine-tuning with 224×224 resolution. ∗ denotes token extended through existing strategies [57, 50, 55]. The best and second-best performances are marked **bold** and underlined.

| | pre-train | | FT | RefCOCO | | | RefCOCO+ | | | RefCOCOg | | |
|---|---|---|---|---|---|---|---|---|---|---|---|---|
| Bridge | tokens | time | tokens | val | test-A | test-B | val | test-A | test-B | val | test | Avg. |
| *224 resolution fine-tuning* | | | | | | | | | | | | |
| Linear | 256 | 0.82× | 256† | 88.37 | 92.38 | 82.36 | 82.81 | 88.82 | 74.31 | 83.37 | 84.82 | 84.66 |
| CoS | 80 | 0.42× | 80† | 85.54 | 90.60 | 79.31 | 79.81 | 87.43 | 71.83 | 81.78 | 82.08 | 82.30 |
| | 80 | 0.42× | 336† | 88.43 | 92.58 | 83.45 | 82.79 | 89.07 | 75.91 | 83.66 | 85.14 | 85.10 |
| *448 resolution fine-tuning* | | | | | | | | | | | | |
| Resamp. | 336 | 1.00× | 336 | 86.46 | 90.84 | 79.82 | 80.66 | 87.74 | 71.63 | 82.25 | 82.64 | 82.76 |
| | 80 | 0.42× | 80 | 83.59 | 89.27 | 76.19 | 77.28 | 84.77 | 67.11 | 78.84 | 79.57 | 79.58 |
| | 80 | 0.42× | 400* | 88.02 | 92.19 | 83.08 | 82.09 | 89.12 | 74.82 | 84.17 | 84.62 | 84.76 |
| CoS | 336 | 1.00× | 336 | 89.20 | 92.96 | 84.73 | 83.83 | 90.57 | 76.97 | 85.48 | 86.26 | 86.25 |
| | 80 | 0.42× | 80 | 86.32 | 91.06 | 81.43 | 80.47 | 87.72 | 74.29 | 82.95 | 82.97 | 83.40 |
| | 80 | 0.42× | 336 | 89.37 | 93.21 | 83.96 | 84.21 | 90.22 | 76.58 | **86.05** | 85.89 | 86.19 |
| | 80 | 0.42× | 1296 | 89.20 | 93.02 | **85.46** | 83.72 | 90.17 | 77.40 | 85.78 | 86.47 | 86.40 |
| *further accelerations* | | | | | | | | | | | | |
| CoS | 48 | 0.35× | 784 | **89.61** | **93.51** | 84.93 | **84.65** | **90.85** | **77.79** | 85.80 | **86.87** | **86.75** |
| | 32 | 0.27× | 336 | 86.97 | 91.11 | 81.18 | 81.30 | 87.83 | 73.72 | 82.58 | 82.84 | 83.44 |
| | 32 | 0.27× | 528 | 88.11 | 92.35 | 83.51 | 83.23 | 89.45 | 75.99 | 84.23 | 84.84 | 85.21 |

Table 4: **Ablations on post-pretrain compound scaling of visual tokens.** G./L. denotes scaling strategy over global tokens (window size=16) and local tokens (window size=4). We report the average performance of captioning, VQA, and text recognition. S-I denotes the image subset of SEEDBench [43]. The best and second-best performances are marked **bold** and underlined.

| | G. | L. | res. | win. sizes | tokens | ∑tokens | Caps | VQA | Text | MMB | POPE | S-I |
|---|---|---|---|---|---|---|---|---|---|---|---|---|
| baseline | - | - | 224 | [16, 4] | [16, 64] | 80 | 111.1 | 66.2 | 84.5 | 68.6 | 84.4 | 64.6 |
| | - | - | 448 | [32, 8] | [16, 64] | 80 | 113.8 | 67.0 | 97.4 | 69.0 | 84.5 | 65.0 |
| Win. Scale | ✓ | × | 224 | [16, 8, 4] | [16, 64, 64] | 144 | 111.3 | 66.0 | 84.8 | 67.9 | 83.6 | 64.8 |
| | × | ✓ | | [16, 2] | [16, 256] | 272 | 112.4 | 67.0 | 86.5 | 68.0 | 84.7 | 66.0 |
| | ✓ | ✓ | | [16, 8, 2] | [16, 64, 256] | 336 | 112.7 | 66.9 | 86.4 | 69.2 | 84.8 | 65.9 |
| Res. Scale | ✓ | × | 448 | [32, 16, 8] | [16, 64, 64] | 144 | 114.6 | 67.0 | 97.7 | 68.4 | 84.8 | 64.9 |
| | × | ✓ | | [32, 4] | [16, 256] | 272 | 115.0 | 67.4 | 100.4 | 69.9 | 86.0 | 66.5 |
| | ✓ | ✓ | | [32, 16, 4] | [16, 64, 256] | 336 | 115.5 | 67.8 | 100.8 | 70.2 | 85.9 | 66.4 |
| Com. Scale | ✓ | × | 448 | [32, 8, 4] | [16, 256, 256] | 528 | 115.4 | 67.8 | 100.2 | 69.7 | 86.2 | 67.0 |
| | × | ✓ | | [32, 16, 2] | [16, 64, 1024] | 1104 | 114.6 | 68.0 | 101.2 | **70.8** | 86.0 | **67.5** |
| | ✓ | ✓ | | [32, 8, 2] | [16, 256, 1024] | 1296 | **115.6** | **68.1** | **101.3** | 70.3 | **86.4** | **67.5** |

**Visual grounding.** Table 3 shows the comparison between Chain-of-Sight and its baselines on referring expression comprehension (REC). The conclusions are consistent with the above findings. Since Chain-of-Sight incorporates both global and local contexts, using our method notably boost the performance of visual resamplers, achieving an improvement of 3.82 and 3.49 on the average performance when using 80 and 336 visual tokens, respectively. Notably, for the REC task, when compared with Chain-of-Sight model pretrained with 336 tokens, we can achieve at most a 2.85× acceleration (reducing the wall-clock training time to 0.35× using 48 tokens for pre-training) without performance loss. Against the standard visual resampler pre-trained with 336 tokens, our 32-token-variant perform favourably, while reducing the training cost by 73%.

**Post-pre-train compound scaling of visual tokens.** In Table 4, we ablate our token scaling strategy. The experiment share the same pre-training, where the baselines are models fine-tuned with 80 visual tokens at $224^2$ and $448^2$ resolutions. For scaling up the number of tokens, we separate the ablations on global and local tokens. Notably, we find scaling up the global tokens alone has negligible effects on the average performances, while scaling up local tokens brings notable improvement on various benchmarks. Combining both further brings slight improvements over the model with up-scaled local tokens on almost all benchmarks. Hence, we use 1296 tokens for fine-tuning our final model.

Table 5: **Comparison with SoTA methods on 10 benchmarks.** Despite that we have only employed LoRA to fine-tune the language model, our model achieves a competitive performance against existing approaches in many benchmarks. *PT tks.* indicates the number of visual tokens used for pre-training and *Parm.* indicates the trainable parameters for the whole model. * indicates at least part of the training set is observed during training. Best performance is marked **bold**. Gray fonts indicate models of larger sizes than ours.

| Model | LLM | PT tks. | Parm. | VQA$^{v2}$ | GQA | VizWiz | SQA$^I$ | VQA$^T$ | POPE | MME | MMB | SEED$^I$ |
|---|---|---|---|---|---|---|---|---|---|---|---|---|
| InstructBLIP-13B [22] | Vicuna-13B | 32 | 188M | - | 49.5 | 33.4 | 63.1 | 50.7 | 78.9 | 1212.8 | - | - |
| LLaVA-1.5-13B [56] | Vicuna-13B | 576 | 13B | 80.0* | 63.3* | 53.6 | 71.6 | 61.3 | 85.9 | 1531.3 | 67.7 | 68.1 |
| CogVLM-17B [97] | Vicuna-7B | 256 | 10B | 82.3* | - | - | 91.2* | 70.4 | 87.9 | - | 77.6 | 72.5 |
| VILA-13B [52] | Vicuna-13B | 576 | 13B | 80.8* | 63.3* | 60.6 | 73.7 | 66.6 | 84.2 | 1570.1 | 70.3 | - |
| Honeybee-13B [10] | Vicuna-13B | 256 | 13B | - | - | - | - | - | 85.5 | 1629/315 | 73.2 | 68.2 |
| Mini-Gemini-13B [48] | Vicuna-13B | 576 | 13B | - | - | - | - | 65.9 | - | 1565/322 | 68.5 | - |
| InstructBLIP-7B [22] | Vicuna-7B | 32 | 188M | - | 49.2 | 34.5 | 60.5 | 50.1 | - | - | 36.0 | 58.8 |
| Shikra [12] | Vicuna-7B | - | 7B | 77.4* | - | - | - | - | - | - | 58.8 | - |
| IDEFICS-9B [41] | - | 64 | 9B | 50.9 | 38.4 | 35.5 | - | 25.9 | - | - | 48.2 | - |
| Qwen-VL [4] | Qwen-7B | 256 | 8B | 78.8* | 59.3* | 35.2 | 67.1 | 63.8 | - | - | 38.2 | 62.3 |
| LLaVA-1.5-7B [56] | Vicuna-7B | 576 | 7B | 78.5* | 62.0* | 50.0 | 66.8 | 58.2 | 85.9 | 1510.7 | 64.3 | - |
| mPLUG-Owl2 [99] | LLaMA2-7B | 64 | 7B | 79.4* | 56.1 | 54.5 | 68.7 | 58.2 | 85.8 | 1450.2 | 64.5 | 57.8 |
| Honeybee-7B [10] | Vicuna-7B | 144 | 7B | - | - | - | - | - | 83.2 | 1584/307 | 70.1 | 64.5 |
| VILA-7B [52] | Vicuna-7B | 576 | 7B | 79.9* | 62.3* | **57.8** | 68.2 | 64.4 | 85.5 | 1533.0 | 68.9 | - |
| Mini-Gemini-7B [48] | Vicuna-7B | 576 | 7B | - | - | - | - | 65.2 | - | 1523/316 | 69.3 | - |
| **CoS-7B** | Vicuna-7B | 80 | 532M | 82.9* | 64.0* | 50.7 | 93.9* | 65.1 | 85.9 | 1549/301 | 72.8 | 68.9 |
| **CoS-8B** | LLaMA3-8B | 80 | 540M | **84.3*** | **65.3*** | - | **95.7*** | 67.6 | **86.9** | **1598/308** | **76.6** | **73.1** |

In terms of training efficiency in the fine-tuning stage, the fastest model (resolution 224 with 80 tokens) is twice as fast as the medium model (resolution 448 with 336 tokens), and uses around 25% of the time spent on training the model with the 1296 visual tokens. However, since the wall-clock time required for supervised fine-tuning is substantially smaller than the pre-training stage, such an increment on the training time during fine-tuning is acceptable.

## 3.3 Comparison with existing approaches

**Visual question answering and vision-language benchmarks.** Table 5 compare the performance of our model with existing approaches. Since the majority of them fine-tunes the whole language model during fine-tuning, the trainable parameters of existing approaches are substantially larger than our approach. Nevertheless, our Chain-of-Sight has achieved competitive performance against existing approaches on many benchmarks, reaching top performance on visual question answering and MMBench among models of the same scale with less than 10% of the trainable parameters. Since the model did not go through an instruction tuning stage, the performance on MME and Vizwiz is not satisfactory. We include the results for more benchmarks in the appendix.

**Visual grounding.** We compare our model with the existing approaches on visual grounding in Table 6. Despite that the only data source of object localization for training our Chain-of-Sight model is the RefCOCO datasets [38, 67, 103], and that our language model is adapted with LoRA [34], our model achieves a leading performance on these three benchmarks, when compared to existing approaches of a similar scale.

## 4 Related work

**Multi-modal large language models.** Since the introduction of the Transformer arhictecture [94] and large-scale pre-training [25, 79], language models have been advancing rapidly [91, 92, 108, 80, 70, 5, 3]. Recently, they are shown to be able to handle various types of data, such as vision [72, 58, 44, 2] and audio [66, 89], leading to a series of multi-modal language models (MLLMs) [4, 12, 99, 109]. The visual capabilities of MLLMs are mainly enabled through transforming visual features into visual tokens, which can be roughly categorized into two types. One uses linear projection to feed image patches into LLMs [58, 11, 16, 93, 97], and the other uses learnable prompts and cross-attentions to aggregate information from the whole feature map [44, 4, 2, 99, 50]. Alternatively, Honeybee [10] proposes a convolutional model for combining the benefit of both. Most existing approaches use an identical number of visual tokens throughout pre-training and fine-tuning. Though some of the recent works have exploited raising the visual tokens during fine-tuning with increased resolution to

Table 6: **Performance comparison on referring expression comprehension** compared with existing approaches. † indicates models fine-tuned with LoRA. Best performance is marked **bold**. Gray fonts indicate models of larger sizes than ours.

| Model | LLM | RefCOCO | | | RefCOCO+ | | | RefCOCOg | | |
| | | val | test-A | test-B | val | test-A | test-B | val | test | Avg. |
|---|---|---|---|---|---|---|---|---|---|---|
| Shikra-13B [12] | Vicuna-13B | 87.83 | 91.11 | 81.81 | 82.89 | 87.79 | 74.41 | 82.64 | 83.16 | 83.95 |
| Ferret-13B [100] | Vicuna-13B | 89.48 | 92.41 | 84.36 | 82.81 | 88.14 | 75.17 | 85.83 | 86.34 | 85.57 |
| Griffon v2 [105] | LLaMA2-13B | 89.60 | 91.80 | 86.50 | 81.90 | 85.50 | 76.20 | 85.90 | 86.00 | 85.42 |
| CogVLM-17B [97] | Vicuna-7B | 92.76 | 94.75 | 88.99 | 88.68 | 92.91 | 83.39 | 89.75 | 90.79 | 90.25 |
| MAttNet [102] | - | 76.40 | 80.43 | 69.28 | 64.93 | 70.26 | 56.00 | 66.67 | 67.01 | 68.87 |
| OFA-L [96] | - | 79.96 | 83.67 | 76.39 | 68.29 | 76.00 | 61.75 | 67.57 | 67.58 | 72.65 |
| UNITER [17] | - | 81.41 | 87.04 | 74.17 | 75.90 | 81.45 | 66.70 | 74.02 | 68.67 | 76.17 |
| MDETR [37] | - | 86.75 | 89.58 | 81.41 | 79.52 | 84.09 | 70.62 | 81.64 | 80.89 | 81.81 |
| Shikra-7B [12] | Vicuna-7B | 87.01 | 90.61 | 80.24 | 81.60 | 87.36 | 72.12 | 82.27 | 82.19 | 82.93 |
| Ferret-7B [100] | Vicuna-7B | 87.49 | 91.35 | 82.45 | 80.78 | 87.38 | 73.14 | 83.93 | 84.76 | 83.91 |
| MiniGPTv2† [11] | LLaMA2-7B | 88.69 | 91.65 | 85.33 | 79.97 | 85.12 | 74.45 | 84.44 | 84.66 | 84.29 |
| Qwen-VL-7B [4] | Vicuna-7B | 89.36 | 92.26 | 85.34 | 83.12 | 88.25 | 77.21 | 85.58 | 85.48 | 85.83 |
| **CoS-7B** † | Vicuna-7B | 90.72 | 93.83 | 85.83 | 86.03 | 91.02 | 78.63 | 87.46 | 87.94 | 87.68 |
| **CoS-8B** † | LLaMA3-8B | **92.67** | **95.14** | **88.89** | **89.12** | **93.63** | **83.25** | **89.56** | **90.42** | **90.33** |

enhance downstream performance [50, 55, 57, 69], the large set of visual tokens for each image still presents a major bottleneck for the pre-training stage.

**Efficient model pre-training.** As the model size consistently expands, the efficiency of training large models has become increasingly important. Beyond efforts in the system optimizations [81, 82, 24, 23], the pre-training of large models can be accelerated by sparse computation, such as masking [46, 77] or mixture of experts [28, 51]. Our approach presents a novel perspective for accelerating pre-training for MLLMs by reducing visual tokens required.

**Multi-scale hierarchy in vision.** Multi-scale hierarchy is a fundamental property in vision, which has led to the introduction and evolution of convolutional networks [30, 42, 40, 33] as well as its application in various vision problems [61, 83, 53, 13]. Recently, transformers are also shown to benefit from multi-scale hierarchy [98, 60, 27, 47]. This work extends multi-scale hierarchy to language models for stronger visual capabilities and higher training efficiency.

## 5  Discussions

**Limitations.** Despite the strong performance and the notable acceleration achieved by Chain-of-Sight, our approach leverages parameter efficient fine-tuning (PEFT) for adapting language models. Hence, the generality of the final model might be limited, compared to approaches that fine-tunes the whole language model during supervised fine-tuning process [57, 10, 52] or even the pre-training stage [63, 4, 26]. This is mainly due to the limited training resources and is exactly what motivates us to explore efficient pre-training methods. We believe the pre-train acceleration achieved by the presented approach has stronger potentials beyond our results.

**Conclusions.** In this work, we set out to accelerate the pre-training phase of MLLMs. Motivated by the unbalance between the number of visual and text tokens during pre-training, we present Chain-of-Sight to reduce the number of token required for pre-training. Chain-of-Sight produces visual tokens of multiple visual scales, providing various level of granularity for the MLLMs to have better perception capabilities. The proposed compound token scaling strategy in the fine-tuning stage can substantially increase the number of tokens post pre-train, such that the model can achieve competitive performance despite the low token count during pre-training. Empirical results have shown that our Chain-of-Sight is capable of achieving a $3.7\times$ speed up in the pre-training process with on-par or better downstream performances. We hope our research can facilitate further investigations in efficient pre-training of MLLMs.

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

## A  Broader impact

The Chain-of-Sight model employs off-the-shelf pre-trained language models. As such, all CoS models inherits the shortcomings from the language model, including certain model biases or hallucinations. While to some extent CoS enhances the visual capabilities of language models, proper assessment and safety precautions are still required before deploying our model.

## B  Details on multi-level feature aggregation

Our multi-scale visual resamplers consider visual hierarchy in two aspects. In addition to the multiple spatial scales detailed in the manuscript, we also enable the visual resamplers to aggregate from multiple feature levels in the visual backbone, which is useful in a wide range of visual tasks [47, 53] but often neglected in current MLLMs.

In Fig. A1, we demonstrate the structure for the multi-level feature aggregation. Essentially, we exploit features from multiple layers in the visual backbone, and the learnable queries aggregate sequentially from lower level to higher level features.

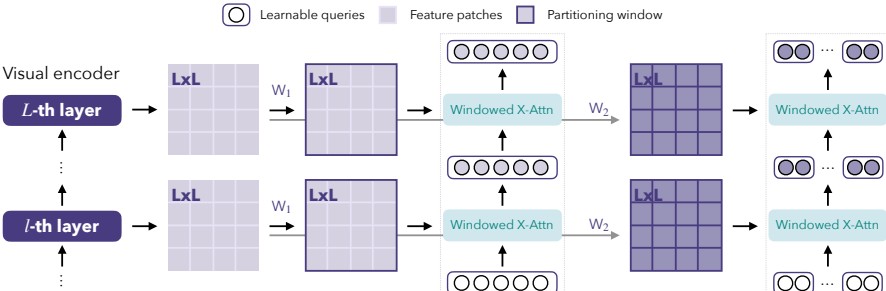

Figure A1: **Multi-level feature aggregation** in the multi-scale visual resamplers of Chain-of-Sight.

## C  Details on training data and evaluation benchmarks

Here, we provide details on the training data and the evaluation benchmarks.

Table A1: **Dataset statistics used for Chain-of-Sight pre-training**. MeanL., 50%L., and 90%L. indicates the mean length, the 50 percentile, and the 90 percentile length of the input text tokens.

| Task | Dataset | MeanL. | 50%L. | 90%L. | Statistical count |
|---|---|---|---|---|---|
| Captioning | COYO [7] | 24.8 | 20 | 44 | 46M |
| | CC3M&CC12M [85] | 22.5 | 17 | 43 | 10M |
| | COCO [54] | 13.4 | 14 | 19 | 0.6M |
| | VG Caption [39] | 7.9 | 7 | 11 | 1M |
| | SBU [75] | 18.2 | 19 | 43 | 0.8M |
| General VQA | VQAv2 [32] | 10.6 | 10 | 14 | 0.6M |
| | GQA [35] | 13.3 | 13 | 19 | 0.9M |
| Knowledge-based VQA | OK-VQA [68] | 13.4 | 13 | 18 | 9k |
| | AOK-VQA [84] | 40.1 | 40 | 46 | 68k |
| | ScienceQA [65] | 45.5 | 38 | 77 | 19k |
| Text | TextVQA [87] | 13.1 | 13 | 17 | 35k |
| | OCRVQA [71] | 15.1 | 13 | 25 | 1M |
| | TextCaps [86] | 18.2 | 17 | 25 | 0.1M |
| REC/REG | RefCOCO [38] RefCOCO+ [103] RefCOCOg [67] | 18.0/7.3 | 18/6 | 19/13 | 321k |

Table A1 shows per-dataset length distributions of our training data. The majority of the data used in the pre-training stage have an average length lower than 25, which provides strong motivation for our approach that reduces the number of visual tokens for pre-training acceleration.

Table A2 shows the detailed description on the benchmarks we use to evaluate our model.

Table A2: Summary of the evaluation benchmarks.

| Task | Dataset | Description | Split | Metrics |
|---|---|---|---|---|
| Captioning | NoCaps [1] | Captioning of natural images. | val | CIDEr (↑) |
| | Flickr [76] | Captioning of natural images. | karpathy-test | CIDEr (↑) |
| | COCO [54] | Captioning of natural images. | karpathy-test | CIDEr (↑) |
| General VQA | VQAv2 [32] | VQA on natural images. | test-dev | VQA Score(↑) |
| | GQA [35] | VQA on scene understanding and reasoning | test-balanced | VQA Score (↑) |
| | OK-VQA [68] | VQA on natural images requiring outside knowledge. | val | VQA Score (↑) |
| | ScienceQA-Img [65] | Multi-choice VQA on a diverse set of science topics | test | Accuracy (↑) |
| Text-rich benchmarks | TextVQA [87] | VQA on natural images containing text. | val | VQA Score (↑) |
| | TextCaps [86] | Captioning of natural images containing text. | test | CIDEr (↑) |
| LVLM Benchmarks | SEED-Bench [43] | Multi-choice VQA on a diverse set of topics | IMG | Accuracy (↑) |
| | MMBench [59] | Multi-choice VQA on a diverse set of topics | test | Accuracy (↑) |
| | MME [29] | Open-ended VL Benchmark by yes/no questions | Perception & Cognition | Accuracy (↑) |
| | POPE [49] | Multi-choice VQA for testing hallucinations | overall | F1-Score (↑) |
| | MMMU [104] | VQA on a diverse set of topics | val | Accuracy (↑) |
| Grounding | RefCOCO [38] | Refer grounding on natural images. | overall | Accuracy (↑) |
| | RefCOCO+ [103] | Refer grounding on natural images. | overall | Accuracy (↑) |
| | RefCOCOg [67] | Refer grounding on natural images. | overall | Accuracy (↑) |

## D  Detailed training settings

We include the detailed parameters for training Chain-of-Sight in Table A3.

Table A3: Training hyperparameters of the chain-of-sight models.

| Configuration | Multi-task pre-training | Supervised fine-tuning |
|---|---|---|
| Image resolution | $224^2$ \| $448^2$ | $448^2$ |
| ViT initialization | CLIP ViT-L/14 | CLIP ViT-L/14 |
| ViT freeze | yes \| no | no |
| LLM adaptation | LoRA (r=64) | LoRA (r=64) |
| Optimizer | AdamW [62] | |
| Optimizer hyperparameter | $\beta_1 = 0.9, \beta_2 = 0.98$ | |
| Peak learning rate | $2e^{-4}$ \| $3e^{-5}$ | $3e^{-5}$ |
| Minimum learning rate | $1e^{-6}$ | $1e^{-6}$ |
| ViT learning rate decay | - \| 0.9 | 0.9 |
| ViT Drop path rate | 0 | |
| Learning rate schedule | cosine decay | |
| Weight decay | 0.1 | |
| Training steps | 120000 \| 30000 | 20000 |
| Warm-up steps | 2000 | 2000 |
| Global batch size | 512 | 256 |
| Numerical precision | `bfloat16` | |

## E  Further results

We provide further empirical results for CoS-7B and CoS-8B in Table A4.

Table A4: Further empirical results. LR: Logic Reasoning, AR: Attribute Reasoning, RR: Relation Reasoning, FP-S: Fine-grained Perception (Single-instance), FP-C: Fine-grained Perception (Cross-instance), CP: Coarse Perception.

| Model | Regular | | | | MMBench | | | | | | | Other |
|---|---|---|---|---|---|---|---|---|---|---|---|---|
| | OK | COCO | NoCaps | Flickr | LR | AR | RR | FP-S | FP-C | CP | Total | MMMU$_v$ |
| **CoS-7B** | 60.3 | 143.0 | 119.7 | 86.0 | 46.6 | 80.9 | 69.5 | 74.1 | 62.2 | 82.7 | 72.8 | 35.4 |
| **CoS-8B** | 62.7 | 142.5 | 119.7 | 85.0 | 46.6 | 82.9 | 80.0 | 77.8 | 71.3 | 84.1 | 76.6 | 39.7 |

