# OpenReview forum: "Accelerating Pre-training of Multimodal LLMs via Chain-of-Sight"
_NeurIPS.cc/2024/Conference — NeurIPS 2024 poster_

### Official Review · Reviewer_Q826 · 2024-07-11

**Soundness:** 4
**Presentation:** 4
**Contribution:** 4
**Rating:** 7
**Confidence:** 5

**Summary:**

This paper introduces Chain-of-Sight (CoS), a novel vision-language bridge module designed to accelerate the pre-training of Multimodal Large Language Models (MLLMs). The key innovation is a sequence of visual resamplers that capture visual details at various spatial scales, allowing for a significant reduction in visual tokens during pre-training while maintaining or improving performance. The authors conduct sufficient experiments, including the evaluation on diverse benchmarks and three scaling experiments, to justify their assumptions and model designs.

**Strengths:**

1. The Chain-of-Sight approach is a novel combination of multi-scale visual processing and token scaling, addressing a critical efficiency bottleneck in MLLM pre-training.
2. Extensive experiments support the claims, demonstrating significant pre-training acceleration (up to 73% reduction in wall-clock time) without sacrificing performance.
3. The paper is well-organized and clearly written.
4. The work addresses a crucial challenge in MLLM development: the computational cost of pre-training.

**Weaknesses:**

One minor weakness is that the authors have not discussed the potential impact of the scale of training data. Multiple compared MLLMs (e.g. Qwen, VILA) in Tables 5 and 6 are probably pre-trained on unequal sizes of data from diverse sources. It will help to clarify how the proposed Chain-of-Sight improves the performance on various tasks if the authors rule out the impact of the differences in the pre-training data.

**Questions:**

Q1: See the Weakness above.

Q2: Will there be differences in the convergence speed of different compound scaling strategies? Which combination of window scale, resolution scale, and compound scale exhibits the fastest convergence?

Q3: Do the authors interpolate the positional embeddings when increasing the input resolution mentioned in Line 175?

Q4: Do the authors mean to refer to Table 3 instead of Table 6 in Line 222?

**Limitations:**

None.

---

> ### Author Rebuttal · Authors · 2024-08-07
>
> **Q1.** Ruling out the impact of the scale of training data.
>
> Thank you for pointing this out. We agree that the scale of training data is one of the most crucial contributing factor in the downstream performance of MLLMs.
>
> However, in our humble opinion, it is hard to single out the impact of the scale of training data on the downstream performances, since the scale of data is entangled with many design choices and hyper-parameter settings in training large models, such as the data composition during supervised fine-tuning, model architecture (the visual backbone and the language model), training strategy (freezing or training the visual backbone and the language model), *etc*.
>
> In order to compare with existing connector structures used in modern MLLMs (e.g., Qwen and VILA), we categorize them into two groups:
>
> - linear projection based MLLMs (e.g., LLaVA, VILA, DeepSeek-VL, *etc*.),
> - and resampler based MLLMs (e.g., Qwen and mPLUG-Owl2).
>
> The downstream performances of these two types of connectors are compared with Chain-of-Sight in Table 2 and Table 3, with identical training configurations, including the training data in pre-training and fine-tuning, the visual backbone, the language model, and the training strategy.
>
> | Method    | time  | # PT/SFT Tks. | Res. | Cap   | VQA | Text | MMB | POPE | SEED-I | Avg performances |
> | ------    | ----  | ------------- | ---- | ---   | --- | ---- | --- | ---- | ------ | ---------------- |
> | Linear    | 0.82x | 256/256       | 224  | 111.8 | 72.4| 84.4 | 67.9| 83.2 | 66.3   | 84.24            |
> | **CoS**   | 0.42x | 80/80         | 224  | 111.1 | 72.3| 84.6 | 68.6| 84.4 | 64.6   | 84.09 (-0.15)    |
> | **CoS**   | 0.42x | 80/336        | 224  | 112.7 | 72.6| 86.4 | 69.2| 84.8 | 65.9   | **85.10** (+0.86)|
>
> | Method    | time  | # PT/SFT Tks. | Res. | Cap   | VQA | Text | MMB | POPE | SEED-I | Avg performances |
> | ------    | ----  | ------------- | ---- | ---   | --- | ---- | --- | ---- | ------ | ---------------- |
> | Resampler | 1.00x | 336/336       | 448  | 113.4 | 71.4| 98.4 | 66.9| 85.3 | 66.2   | 86.76            |
> | **CoS**   | 0.42x | 80/336        | 448  | 115.5 | 73.3| 100.8| 70.2| 85.9 | 66.4   | 88.63 (+1.87)    |
> | **CoS**   | 0.42x | 80/1296       | 448  | 115.7 | 74.0| 101.3| 70.3| 86.4 | 67.5   | **89.13** (+2.37)|
>
> *Note 1: The number of VQA here is different from the ones in Table 4 as the VQA in Table 4 did not take ScienceQA into account.*
>
> *Note 2: The average performances of Captioning, VQA, and Text, as well as the overall average performances are calculated based on the results in Table 2.*
>
> From the above tables, we can observe that under the same configurations, Chain-of-Sight outperforms both linear projection and resampler based methods.
>
> **Q2.** Differences in the convergence speed of different compound scaling strategies.
>
> Counter-intuitively, we find different compound scaling strategies lead to a similar convergence speeds during the fine-tuning process, despite the huge differences in the downstream performances. Nevertheless, the model with more visual tokens and higher resolutions generally achieve a lower loss throughout the fine-tuning process.
>
> **Q3.** Interpolation of positional embeddings whtn increasing the input resolution.
>
> Yes, the positional embeddings in the pre-trained CLIP backbone are interpolated when we increase the input resolution to 448.
>
> **Q4.** Refering to Table 3 instead of Table 6 in Line 222.
>
> Yes, we meant to refer to Table 3 in Line 222. We will correct this in our revisions.

---

> > ### Comment · Reviewer_Q826 · 2024-08-13
> > **Response to the authors' rebuttal**
> >
> > Thank you for providing the detailed answers. My concerns have been resolved and the rating of 7 is maintained. Good luck!

---

### Official Review · Reviewer_cV6y · 2024-07-12

**Soundness:** 3
**Presentation:** 3
**Contribution:** 3
**Rating:** 6
**Confidence:** 3

**Summary:**

This paper proposed a post-pretrain token scaling strategy, Chain-of-Sight, to accelerate the pre-training of Multimodal Large Language Models (MLLMs).
Through the proposed method, the authors were able to achieve a significant improvement in pre-training speed.
The authors confirmed through an ablation study that the proposed method exhibits performance similar to existing approaches.

**Strengths:**

The paper is well-written, clearly and coherently presenting ideas.

The proposed method demonstrates performance similar to or better than existing methods, despite having shorter training times.

An ablation study was conducted across various tasks and settings.

**Weaknesses:**

The paper only includes experimental results using CLIP-ViT-L/14 as the visual encoder and Vicuna. Experimental results based on other models are needed.

The explanation of how the number of tokens during pre-training and fine-tuning (e.g., PT:80 FT: 1296) was determined is insufficient.

**Questions:**

see weakness.

**Limitations:**

The authors addressed both the limitations and potential negative social impacts in Section 5 (Conclusion) and Section A (Broader impact).

---

> ### Author Rebuttal · Authors · 2024-08-07
>
> **Q1.** Experimental results based on other models.
>
> Thank you for the constructive comment. We provide experimental results based on different language models in the following table, which shows Chain-of-Sight benefits from stronger language models.
>
> | Language model | # PT/SFT tks. | Caps  | VQA | Text | MMB | POPE | S-I | Avg performance |
> | -------------- | ------------- | ----  | --- | ---- | --- | ---- | --- | --------------- |
> | vicuna-v1.3    | 80/336        | 115.5 | 73.3| 100.8| 70.2| 85.9 | 66.4 | 88.63           |
> | vicuna-v1.5    | 80/336        | 115.0 | 73.8| 99.9 | 70.9| 85.4 | 67.2 | 88.68           |
> | llama3         | 80/336        | 114.1 | 75.8| 101.9| 75.7| 85.6 | 72.4 | 90.26           |
>
> *Note 1: The number of VQA here is different from the ones in Table 4 as the VQA in Table 4 did not take ScienceQA into account.*
>
> *Note 2: The average performances of Captioning, VQA, and Text, as well as the overall average performances are calculated based on the results in Table 2.*
>
> To validate the effectiveness of Chain-of-Sight, we also performed ablation studies with Llama-3-8B, where various nubmers of tokens are used during supervised fine-tuning. The table below shows that the compound scaling strategy effectively improves the downstream performance. Since the pre-training takes long for resampler-based (336 tokens) and linear-based models (256 tokens), we only validate the effectiveness of the compound scaling here, and leave the baseline results to our revisions.
>
> | Language model | # PT/SFT tks. | Caps  | VQA | Text | MMB | POPE | S-I | Avg performance |
> | -------------- | ------------- | ----  | --- | ---- | --- | ---- | --- | --------------- |
> | llama3         | 80/80         | 113.1 | 75.0 | 100.2| 74.3| 85.5 | 70.3 | 89.14           |
> | llama3         | 80/336        | 114.1 | 75.8 | 101.9| 75.7| 85.6 | 72.4 | 90.26           |
> | llama3         | 80/1296       | 114.7 | 76.3 | 103.9| 76.4| 86.6 | 73.5 | 91.11           |
>
> In addition, we also go through the whole training process with different language models, which includes the stage 1 pre-training, high resolution post pre-training, and multi-task supervised fine-tuning.
>
> | Language model | # PT/SFT tks. | VQAv2 | GQA | SQA_I | TextVQA | POPE | MME | MMB | SEED_I | MMMU |
> | -------------- | ------------- | ----- | --- | ----- | ------- | ---- | --- | --- | ------ | ---- |
> | vicuna-v1.3    | 80/1296       | 82.9  | 63.2| 91.6  | 65.3    | 85.0 | 1474/264 | 72.5 | 67.5 | 34.1 |
> | vicuna-v1.5    | 80/1296       | 82.9  | 64.0| 93.9  | 65.1    | 85.9 | 1549/301 | 72.8 | 68.9 | 35.4 |
> | llama3         | 80/1296       | 84.3  | 65.3| 95.7  | 67.6    | 86.9 | 1598/308 | 76.6 | 73.1 | 39.7 |
>
> We will include results based on other language models and visual backbones in our revisions.
>
> **Q2.** Explanation on how the numbers of tokens during pre-training and fine-tuning are determined.
>
> Because of the size of a large model and the scale of training data, it would be infeasible to do a grid search on the number of tokens. Therefore, the model hyperparameters including the window sizes and the number of visual tokens are determined under several key rationals.
>
> *Pre-training.*
>
> In the pre-training stage, we employ the resolution of 224, which gives us the feature of size 16x16.
>
> *Window sizes.* Since the main objective of Chain-of-Sight is to accelerate pre-training by reducing the number of tokens during pre-training, we limit the number of visual scale hierachies to two, *i.e.*, one global view (window size 16) and one local view (window size 4).
>
> *Token counts.* Features with larger window sizes are usually more informative w.r.t. image contents, thus requiring more visual tokens to represent. For efficiency, we intuitively selected 16 visual tokens for encoding the global view (instead of 32 as used in BLIP-2 or more). As for the token counts in the local view, we have experimented with 1, 2, and 4 per window, as in Table 2 and 3. Eventually, we use 4 tokens per window for the local view, which strikes a good balance between the training speed and downstream performance.
>
> *Supervised fine-tuning.*
>
> As mentioned in the manuscript, given a pre-trained Chain-of-Sight model, there are two ways of scaling up the number of tokens, *i.e.,* increasing the resolution and reducing the window sizes, respectively. Both methods can increase the token counts by 4 times for a specific pre-trained window size, producing a 16x token count for each window size, where 80 tokens are scaled up to 1280 tokens. In addition, we keep a copy of 16 global tokens for providing a comprehensive overview of the input image, resulting in a total number of 1296 tokens.
>
> We provide a detailed calculation of the number of tokens in both stages in the table below.
> Detailed explanation on the choice of model hyperparameters will be included in our revisions.
>
> | Stage | Feature size | Win size | # windows | # tks. per win | # tks. in total |
> | - | -- | - | - | -- | - |
> | Pretrain | 16x16 | 16x16 | 1| 16 | 16|
> |  |  | 4x4 | 16 | 4 | 64 |
> |  |  |  |  | Pretrain total | 16+64=80 |
> | Fine-tune | 32x32 | 32x32 | 1| 16 | 16|
> |  |  | 8x8 | 16 | 16 | 256 |
> |  |  | 2x2 | 256 | 4 | 1024 |
> |  |  |  |  | Fine-tune total | 16+256+1024=1296 |

---

### Official Review · Reviewer_TSJ3 · 2024-07-13

**Soundness:** 2
**Presentation:** 2
**Contribution:** 2
**Rating:** 5
**Confidence:** 4

**Summary:**

1. This work proposes Chain-of-Sight, a training method of MLLM that leverages global and local visual contexts effectively.
2. To boost efficiency in the pretraining stage, the authors propose a post-pretrain token scaling strategy. During the pretraining state, it requires significantly fewer visual tokens and cuts down the wall-clock training time by 73% in the pretraining stage. To effectively increase tokens during fine-tuning stage to enhance performance, they propose a compound strategy by manipulating input resolution and window size at the same time.
3. When both use the same number of tokens during fine-tuning, the results achieved by the Chain-of-Sight model pre-trained with 32 tokens match or surpass those obtained using 336 visual tokens throughout the training process.

**Strengths:**

1. The writing is mostly clear and easy to follow, except for the compound scaling part where a clearer illustration is suggested.
2. Comprehensive ablation study. Variations of pretraining token number / pretraining strategy(CoS, Resample) / FT are compared on various tasks including image captioning, visual question answering, text recognition, referring expression comprehension, etc.
3. According to Table 5, CoS-7B achieves good results on VQA benchmarks compared to 7B-level baselines.

**Weaknesses:**

1. The method of hierarchical (multi-scale) visual input has already been explored in many former works, which is contradictory to the expression in L91-92 "Despite this, the potential for harnessing multi-scale visual hierarchies remains under-explored in the context of MLLMs". For example, LLaVA-NeXT [1] and InternLM-XComposer2-4KHD [2] both adopted multi-scale strategies and proved their effectiveness.

2. There are mistakes in the table of experiment results. For example, in Table 3, RefCOCO+, test-A, the second-best one should be 90.57.



[1] Liu, Haotian, et al.  "LLaVA-NeXT: Improved reasoning, OCR, and world knowledge." https://llava-vl.github.io/blog/2024-01-30-llava-next/

[2] Dong, Xiaoyi, et al. "Internlm-xcomposer2-4khd: A pioneering large vision-language model handling resolutions from 336 pixels to 4k hd." *arXiv preprint arXiv:2404.06512* (2024).

**Questions:**

1. In Table 2 and Table 3, why are many of the results of "CoS, pre-train 80, FT 336" better than those of "CoS, pre-train 336, FT 336"? Analysis or explanations are suggested.
2. The settings of ablation experiments are not explained. Is that the same as sec 3.1 training settings? If so, why do the results in Table 5 not agree with those in Table 2&3&4?

**Limitations:**

The experiment is only conducted on 7B-level model with PEFT. Larger scale model with full-parameter training is suggested to prove its effectiveness and extensibility in modern pretraining.

---

> ### Author Rebuttal · Authors · 2024-08-07
>
> **Q1.** The method of hierarchical (multi-scale) visual input has already been explored in many former works, such as LLaVA-NeXT and InternLM-XComposer2-4KHD.
>
> The multi-scale nature is the fundamental characteristic of images manifested by David Marr's pioneering work on vision perception back in 1970s. How to exploit the multi-scale nature effectively has been a key research topic in most vision tasks, if not in all tasks, for half a century, which is likely to continue to be an active research topic in future decades.
>
> Thanks for raising this point and we are happy to discuss the differences in the multi-scale strategy between Chain-of-Sight and the methods used in LLaVA-NeXT and InternLM-XComposer-4KHD, in the following three aspects.
>
> *Conceptual idea*
>
> Both LLaVA-NeXT and InternLM-XComposer-4KHD process the high-resolution input image in two paths. One splits the high-resolution image into partitions of sub-images with a 'base resolution' supported by the backbone, which is 336 for both methods. The other resizes the high-resolution image to 'base resolution' to be processed by the backbone. The multi-scale idea is exploited on the input side before the visual backbone, which is similar to the multi-resolution/multiview strategy [1,2,3].
>
> Differently, inspired by the pyramid structure in contemporary vision models [4,5,6], our Chain-of-Sight method constructs in-model multi-scale features, where multi-scale visual prompts are generated after the visual backbone based on the features of a single 'base-resolution' image. Hence, our method of leveraging multi-scale hierarchy is in parallel with the multi-view method used in LLaVA-NeXT and InternLM-XComposer-4KHD, and it is possible to combine both methods to achieve even stronger visual capabilities.
>
> *Motivation*
>
> The motivations of LLaVA-NeXT and InternLM-XComposer-4KHD are to enable MLLMs for capturing more details in the high-resolution images, while Chain-of-Sight is proposed to leverage the flexiblity of our multi-scale visual resampler such that the pre-training could be accelerated with lower nubmer of visual tokens without compromising performance.
>
> *Methodological details*
>
> In terms of the methodological details, both LLaVA-NeXT and InternLM-XComposer-4KHD leverages a two-level visual hierarchy, *i.e.,* the global and local views. Though Chain-of-Sight splits the input image into global and local windows in the pre-training, it is able to be extended to three or more scale levels during fine-tuning thanks to the compound scaling strategy of Chain-of-Sight.
>
> In addition, we would like to highlight that, as mentioned in Sec 2.3, LLaVA-NeXT and InternLM-XComposer-4KHD can be considered as a special case of our compound scaling, where only the resolution of the input image is scaled up, while the window size is kept the same.
>
> More details will be provided in our revisions.
>
> [1] Karpathy, Andrej, et al. "Large-scale video classification with convolutional neural networks." In CVPR 2014.
>
> [2] Yan, Shen, et al. "Multiview transformers for video recognition." In CVPR 2022.
>
> [3] Feichtenhofer, Christoph, et al. "Slowfast networks for video recognition." In ICCV 2019.
>
> [4] Lin, Tsung-Yi, et al. "Feature pyramid networks for object detection." In CVPR 2017.
>
> [5] Wang, Wenhai, et al. "Pyramid vision transformer: A versatile backbone for dense prediction without convolutions." In ICCV 2021.
>
> [6] Liu, Ze, et al. "Swin transformer: Hierarchical vision transformer using shifted windows." In ICCV 2021.
>
> **Q2.** Mistakes in the experiment section.
>
> Thank you. We will re-examine the manuscript and correct the mistakes.
>
> **Q3.** Why is "CoS, PT80, FT336" stronger than "CoS, PT336, FT336"?
>
> It is indeed the case. Though the performances of the two variants are similar on referring expression comprehension, "CoS, PT336, FT336" underperforms "CoS, PT80, FT336" in almost all the other aspects.
>
> | Model | Caps | VQA | Text | MMB | POPE | S-I | Avg (Table 2) | Avg (Table 3) |
> | ----- | ---- | --- | ---- | --- | ---- | ------ | ------------- | ------------- |
> | CoS, PT80, FT336 | 115.5 | 73.3 | 100.8 | 70.2 | 85.9 | 66.4 | 88.6 | 86.19   |
> | CoS, PT336, FT336 | 113.9 | 72.8 | 97.1 | 68.7 | 84.2 | 66.9 | 87.2 | 86.25   |
>
> *Note 1: The number of VQA here is different from the ones in Table 4 as the VQA in Table 4 did not take ScienceQA into account.*
>
> *Note 2: The average performances of different tasks and the overall average performances are calculated based on the results in Table 2.*
>
> We believe the reason behind this is that the low capacity of the "CoS, PT80" model during pre-training acts as a filtering mechamism for the noisy data, which allows it to learn the more commonly existing distributions in the pre-training data. Specifically, we find the pre-training loss of "CoS, PT336" lower than "CoS, PT80", while the fine-tuning loss of "CoS, PT336, FT336" is notably higher than "CoS, PT80, FT336". Given the higher level of noise in the pre-training data, we believe the higher capacity of "CoS, PT336" model makes it learn more low-quality data than the "CoS, PT80" model, leading to the worse downstream performance of the "CoS, PT336, FT336" model. In fact, similar phenomenon can be observed with resampler-based model. As in Table 2, the average performance of "Resampler, PT80, FT 400" is also 0.6 higher than "Resampler, PT336, FT336".
>
>
> **Q4.** The results in Table 5 is different from that in Table 2&3&4.
>
> Yes, the training settings are different between our final model and the models in the ablation studies.
> The model in Table 5 is trained with an additional post-pre-train high-resolution stage, while for ablations, we skip this stage for efficiency, as mentioned in L184. We will make this clearer in our revision.
>
> **Q5.** Larger scale model with full-parameter training.
>
> Limited by time, we are unable to finish training large-scale model with full-parameter training. We will include the results in our revisions.

---

> > ### Comment · Reviewer_TSJ3 · 2024-08-13
> >
> > Thanks for your reply, my concerns have been partially solved. However, I'm still concerned about the limited technical novelty (Q1), and the effect on larger-scale model or full-parameter training. Therefore, I would raise my score a bit but still be on the borderline (Please consider my score to be 4.5).

---

### Author Rebuttal · Authors · 2024-08-07

We genuinely appreciate the reviewers for dedicating their time and effort to review our manuscript and providing valuable comments and insights. We are encouraged by the reviewers' assessment that

1. Our work addresses a crucial challenge in MLLM development: the computational cost of pre-training (Q826).

2. Our proposed method, Chain-of-Sight

    - is a novel combination of multi-scale visual processing and token scaling, addressing a critical efficiency bottleneck in MLLM pre-training (Q826).
    - reduces the wall-clock training time by 73% for MLLMs without sacrificing performance (TSJ3 and Q826).
    - demonstrates performance similar to or better than existing methods (TSJ3 and cV6y).

2. Our experiments are comprehensive and extensive, which are able to support our claims (TSJ3, cV6y, and Q826).

3. The writing is easy to follow (TSJ3). The manuscript is well-written (cV6y) and well-organized (Q826).

We are committed to addressing the limitations and concerns raised by the reviewers within the specified time frame. We believe that doing so will significantly improve the quality of our manuscript. Below, you will find detailed responses to the questions and comments raised by each reviewer.

---

### Decision · Program_Chairs · 2024-09-25

**Decision:**

Accept (poster)

**Comment:**

This work leverages multi-scale tokens to reduce the number of visual tokens in pre-training VLMs, speeding up the pre-training of VLMs by ~2.8x (including SFT) -- 3.7x (excluding SFT), while maintaining/improving the performance. The approach is validated for a number of models, tasks, and datasets.

The reviewers agreed that this work is clearly presented, and appreciate the gains in pre-training efficiency.
There are concerns regarding no results with bigger models, however, 7B models used in the experiments are of respectable size.

I can recommend this paper for publication given the scope for adoption of the proposed approach due to pre-training efficiency gains.